# Moss bugs shed light on the evolution of complex bioacoustic systems

**Leonidas-Romanos Davranoglou**[1]*, **Viktor Hartung**[2,3]

**1** Oxford University Museum of Natural History, University of Oxford, Oxford, United Kingdom, **2** LWL-Museum of Natural History, Westphalian State Museum with Planetarium, Münster, Germany, **3** Museum für Naturkunde—Leibniz Institute for Evolution and Biodiversity Science, Berlin, Germany

* leonidas-romanos.davranoglou@oum.ox.ac.uk

## Abstract

Vibroacoustic signalling is one of the dominant strategies of animal communication, especially in small invertebrates. Among insects, the order Hemiptera displays a staggering diversity of vibroacoustic organs and is renowned for possessing biomechanically complex elastic recoil devices such as tymbals and snapping organs that enable robust vibrational communication. However, our understanding of the evolution of hemipteran elastic recoil devices is hindered by the absence of relevant data in the phylogenetically important group known as moss bugs (Coleorrhyncha), which produce substrate-borne vibrations through an unknown mechanism. In the present work, we reveal the functional morphology of the moss bug vibrational mechanism and study its presence across Coleorrhyncha and in extinct fossilised relatives. We incorporate the anatomical features of the moss bug vibrational mechanism in a phylogeny of Hemiptera, which supports either a sister-group relationship to Heteroptera, or a sister-group relationship with the Auchenorrhyncha. Regardless of topology, we propose that simple abdominal vibration was present at the root of Euhemiptera, and arose 350 million years ago, suggesting that this mode of signalling is among the most ancient in the animal kingdom. Therefore, the most parsimonious explanation for the origins of complex elastic recoil devices is that they represent secondary developments that arose exclusively in the Auchenorrhyncha.

## Introduction

Communication using substrate-borne vibrations is one of the dominant signalling modalities among animals, being used by at least 200,000 species [1–3]. Vibrational signalling is particularly widespread in small invertebrates, especially insects, which they use in an array of behavioural contexts, such as courtship [4–6], competition [7,8], the transmission of alarm and defensive signals [9–11], and even to trigger synchronous egg-hatching [12]. Among them, true bugs (Hemiptera), a megadiverse insect group with over 107.000 described species [13–15], display the greatest diversity of vibroacoustic organs in the animal kingdom [16]. Stink bugs and their allies (Heteroptera) use abdominal vibration (tremulation) [3], tapping [17,18], wing buzzing [19], and stridulation [16] to generate simple substrate borne vibrations.

**Data Availability Statement:** All evidence used to support our conclusions is provided within the manuscript. Full tomographic data cannot be shared publicly because they are embargoed until 2026 when our research on them will be finalised.

We will then submit them to online databases, such as CXIDB.

**Funding:** This publication arises from research funded by the Leverhulme Trust Early Career Fellowship grant (ECF-2021-199) and the John Fell Oxford University Press Research Fund to L.-R. Davranoglou. V. Hartung was funded by the Elsa Neumann doctorate grant (application number H49023) and the German Academic Exchange travel grant (processing number D/09/04219). Neither of the funders had a role in study design, data collection and analysis, decision to publish, or preparation of the manuscript.

**Competing interests:** The authors have declared that no competing interests exist.

Conversely, various groups of the suborder Auchenorrhyncha employ specialised elastic recoil devices to generate complex vibroacoustic signals that would be impossible to achieve with muscle action alone. The most widely used elastic recoil devices in Auchenorrhyncha involve the buckling of drum-like tymbals in cicadas and their relatives (Cicadomorpha) [20,21] and the recently discovered snapping organs in planthoppers (Fulgoromorpha) [22]. Unsurprisingly, the evolutionary origins of complex traits such as hemipteran vibroacoustic elastic recoil mechanisms have remained a matter of debate [21–24]. Their intricate and microscopic morphologies render them challenging to study, their extreme diversity makes homologizing structures between taxa very difficult, their vibroacoustic organs rarely fossilise, and the systematic placement of certain groups has remained inconclusive, thereby hindering our understanding of how hemipteran biomechanics has changed across time. An elusive group of hemipterans known as the moss bugs (Coleorrhyncha: Peloridiidae) may offer additional insights on the character acquisition that led to the diversity of Hemipteran vibroacoustic mechanisms that we observe today.

Coleorrhyncha (Fig 1) are a relict group that comprises a single recent family, the Peloridiidae, with 37 small (normally 2–3 mm), cryptically coloured, mostly wingless species that feed on bryophytes, and are rarely collected [25–27].

While Peloridiidae are characterised by a Gondwanan distribution, being found in Australia, Lord Howe Island, New Caledonia, New Zealand and southern South America [26–30], their putative extinct relatives that date from the Permian to Cretaceous periods, have been found in Argentina, Australia and Eurasia [31–40]. Coleorrhyncha are characterised by the following suite of unique adaptations: to cope with their deficient diet, Peloridiidae possess biosynthetic bacterial endosymbionts that cover their nutrient requirements [41,42]; their body surface supports a plastron that allows them to stay submerged underwater, a useful trait when living in moist bryophytes [43]; and they perform high-speed jumps (1.5 m s$^{-1}$) to evade predators [44]. At least one species of Peloridiidae, *Hackeriella veitchi* (Hacker, 1932), has been shown to communicate with low frequency (82 Hz) vibrational signals, which are thought to be produced by a tymbal mechanism [45], although this has not been confirmed experimentally. Indeed, the morphology and biomechanics of coleorrhynchan vibrational organs have remained poorly known, which is a major gap in our understanding of the evolution of hemipteran vibrational organs. The most intriguing question here is whether Peloridiidae (and Coleorrhyncha) use complex elastic recoil devices or simple abdominal vibration. Furthermore, the behavioural context of the emitted signals is not known [45].

The phylogenetic relationships within Euhemiptera (= Auchenorrhyncha + Coleorrhyncha + Heteroptera) have remained controversial [46–48] (Fig 2). Some morphological and molecular studies support Coleorrhyncha as the sister group to Heteroptera, jointly forming the suborder known as the Heteropterodea (Fig 2A) [46,47,49–51], while others recover Coleorrhyncha as sister to Auchenorrhyncha (Fig 2B), rendering Heteropterodea paraphyletic [52–54]. Based on the above, the elusive relationships of Coleorrhyncha, coupled with the unknown morphology and biomechanics of their vibrational mechanism, prevent us from understanding how complex character states, such the elastic recoil devices that define Auchenorrhyncha, evolved from simpler precursors.

To address this knowledge gap, we use state-of-the-art synchrotron X-ray microtomography, scanning electron microscopy, and laser Doppler vibrometry to document the morphology and preliminary biomechanics of coleorrhynchan vibrational organs. Through the examination of fossil Coleorrhyncha, we present indications of possible morphological conservatism in moss bug vibrational organs, which may have remained largely unchanged at least since the Late Jurassic (160 mya). To reconstruct the evolution of vibroacoustic mechanisms of Euhemiptera, we integrate the newly acquired morphological data into a morphological matrix

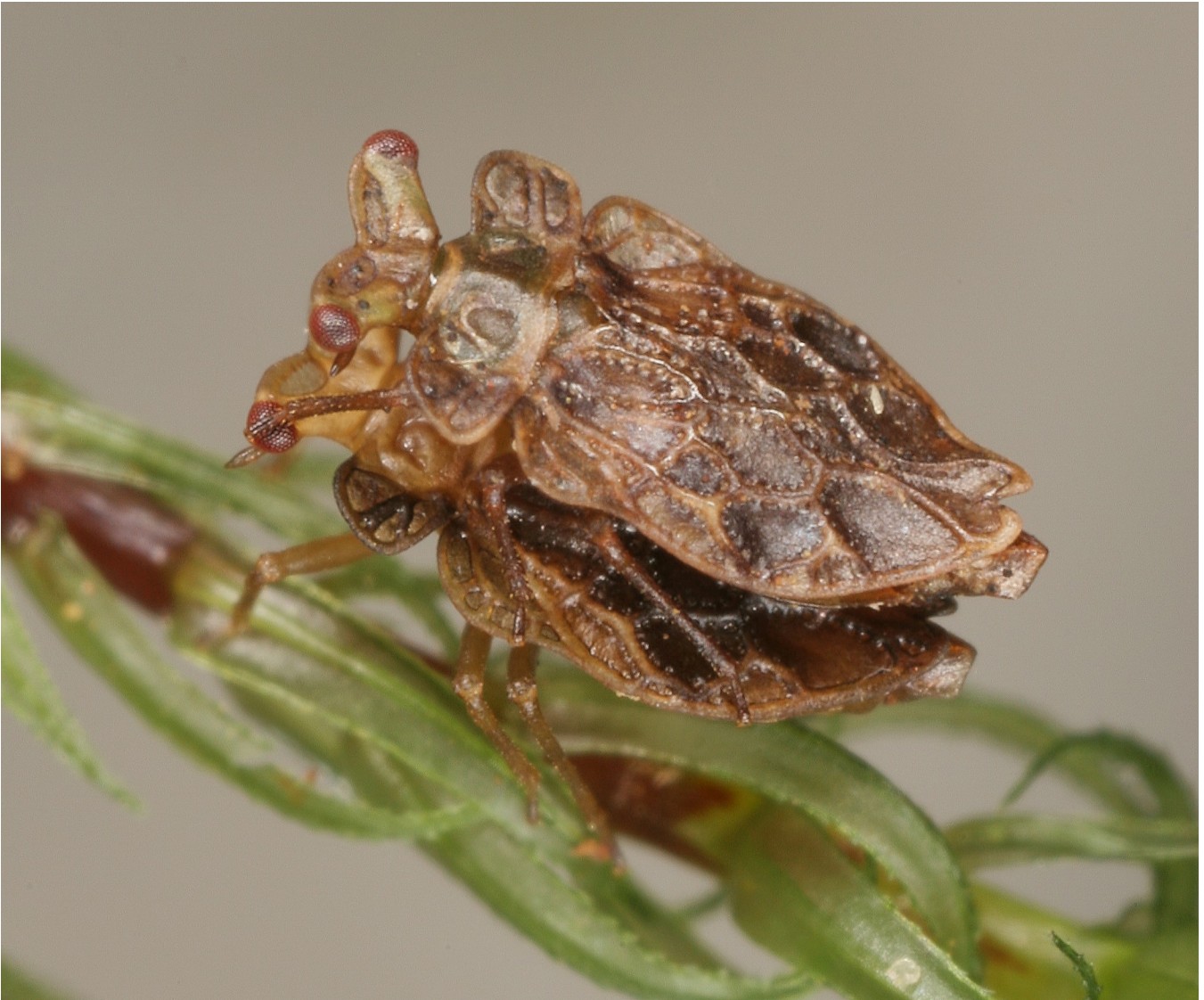

**Fig 1. Two males of *Hackeriella veitchi*, showing the habitus of a generalised moss bug.** (photograph courtesy of Jürgen Deckert).

comprising 111 characters. We show that complex elastic recoil devices evolved at the root of Auchenorrhyncha, whereas simple abdominal vibration likely represents the plesiomorphic state in Euhemiptera.

## Results

### Morphology of the moss bug pregenital abdomen

To characterise the morphology of the coleorrhynchan putative abdominal vibrational organ, we examined the pregenital abdomen of five species, covering a considerable proportion of peloridiid diversity (S1 Table). The following description is based on the combined observations on all these species. As in other Euhemiptera, the putative vibrational organ spans the first two abdominal segments [21,22], which are largely membranous to confer flexibility, except for two central sclerotised plates formed by tergites I-II, respectively (Fig 3A, 3B and

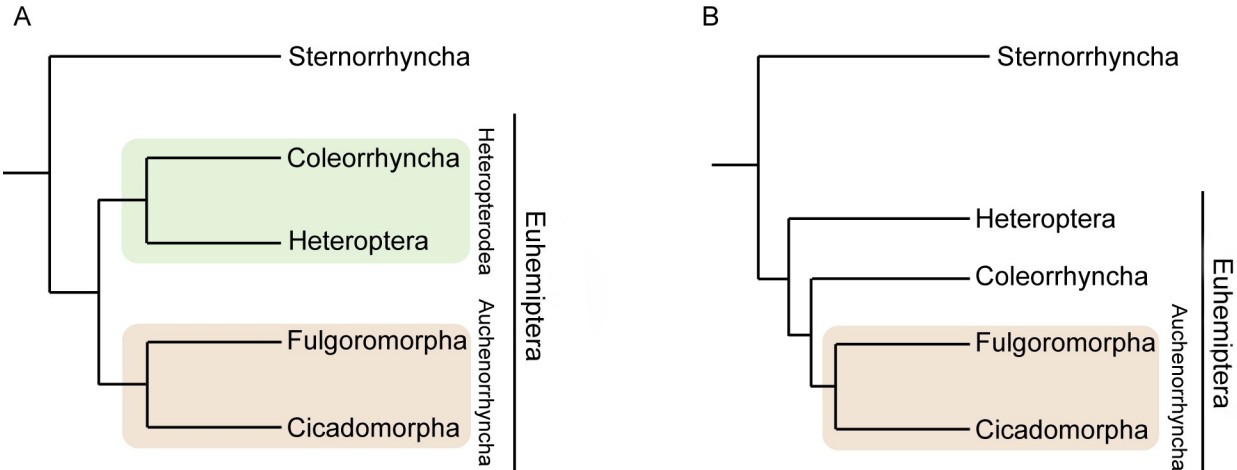

**Fig 2. Some of the previous topologies of the moss bug (Coleorrhyncha) position on hemipteran phylogeny.** A) A sister-group relationship to Heteroptera (forming the lineage Heteropterodea), as suggested by [46,47,49–51]; B) alternative topology, where Coleorrhyncha represent the sister-group to Auchenorrhyncha (based on [52–54]).

3D). Externally visible elastic recoil devices such as a tymbal or a snapping organ are not present.

Tergite I is long but narrow in most peloridiids from Australia and South America (Figs 3A, 3B, 3D, 4A–4C, 5A and 5B), whereas in certain genera (*Oiophysa*, *Xenophyes*) from New Zealand it is short and broad (Figs 4D–4F and 6A). The proportions of the first abdominal tergite may be phylogenetically and biomechanically informative. In all Peloridiidae examined, a distinct posterolateral ridge (PLR) is present on tergite I, whose shape is species-specific (Fig 4).

The form and arrangement of the posterolateral ridge co-varies with the dimensions of tergite I: in longer and narrower tergites there are two lateral folds that join the posterior apodeme in an obtuse angle (Figs 3A, 3B, 3D, 4A–4C, 5A and 5B), whereas broader and shorter tergites possess a posterolateral ridge that is confined only to the posterior margin of tergite I, bordering the antecosta (Figs 4D–4F and 6A). Spiracle I is placed on a semilunar-shaped sclerite that is fused to the metapostnotum (Fig 3B). Tergite II is reduced to a narrow strip in all Coleorrhyncha examined (Figs 3A, 3B, 3D, 4, 5A and 5B) and exhibits no significant variation. Spiracle II is always located on the anterolateral margin of tergite II (Figs 3A, 3B, 3D, 4, 5A 5B and 6A), a condition similar to that of Auchenorrhyncha [21,22]. All remaining spiracles are located ventrally on the connexivum (laterotergites) (Figs 3C, 5D and 6C). Although mentioned in previous studies as absent [30], a ring-like sternite I was found in all species studied (Figs 3C, 6C and 6E). Sternite I is absent from most groups of Heteroptera apart from some Enicocephalomorpha [55] and Gerromorpha [56] (as a secondary reversal) but is present in all Auchenorrhyncha [21,22]. The posterior margin of sternite I is provided with a pair of apodemes which act as attachment sites for muscles (Fig 7A, apo I). Sternite II is strongly constricted medially and expanded laterally (Fig 7A). The border between sternites II-III bears a pair of very large apodemes (Fig 7A, apo II), which have been previously interpreted as part of a tymbal mechanism [23]. Similar apodemes are present in Auchenorrhyncha [20–23], although their homology to those of the Coleorrhyncha is unclear. The remaining pregenital tergites and sternites are morphologically unspecialised. A distinct carina is present on sternites III-V (Fig 3C, crn).

Eleven pairs of muscles are directly associated with the moss bug vibrational mechanism (Fig 7), comprising two pairs of dorsal longitudinal muscles (DLMs), two pairs of ventral

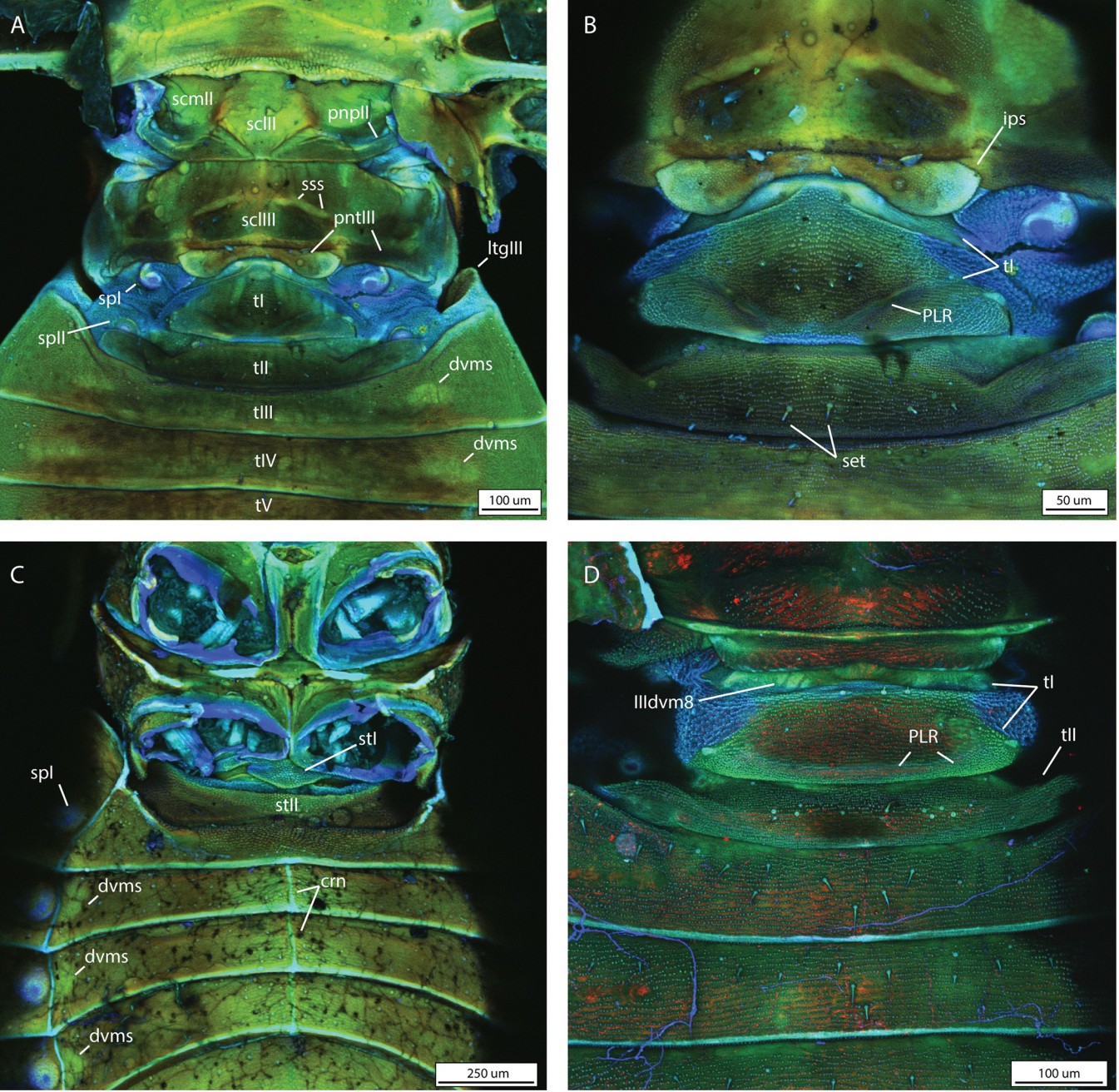

**Fig 3. External morphology of the moss bug pregenital abdomen. including the vibrational organ, as visualised by confocal laser scanning microscopy.**
A) Dorsal view of thorax and proximal pregenital abdomen of *Hemiodoecus leai*; B) same, magnified view; C) ventral view of thorax and pregenital abdomen of *H. leai*; D) dorsal view of proximal pregenital abdomen of *Xenophyes cascus*. Abbreviations: crn = median carina; dvms = dorsoventral muscles; ips = intrapostnotal suture; ltg = laterotergite; PLR = posterolateral ridge; pnp = posterior notal wing process; pnt = postnotum; scl = scutellum; scm = scutum; set = setae; sp = spiracle; sss = scuto-scutellar suture; st = sternite; t = tergite.

longitudinal muscles (VLMs), and six pairs of dorsoventral muscles (DVMs) (for a list of the origins and attachments of pregenital musculature, refer to S2 Table). Unlike the muscles that operate the elastic recoil devices of other Hemiptera [20–23], none of the moss bug pregenital muscles display hypertrophy that would enable high power motions (Fig 7A). The largest

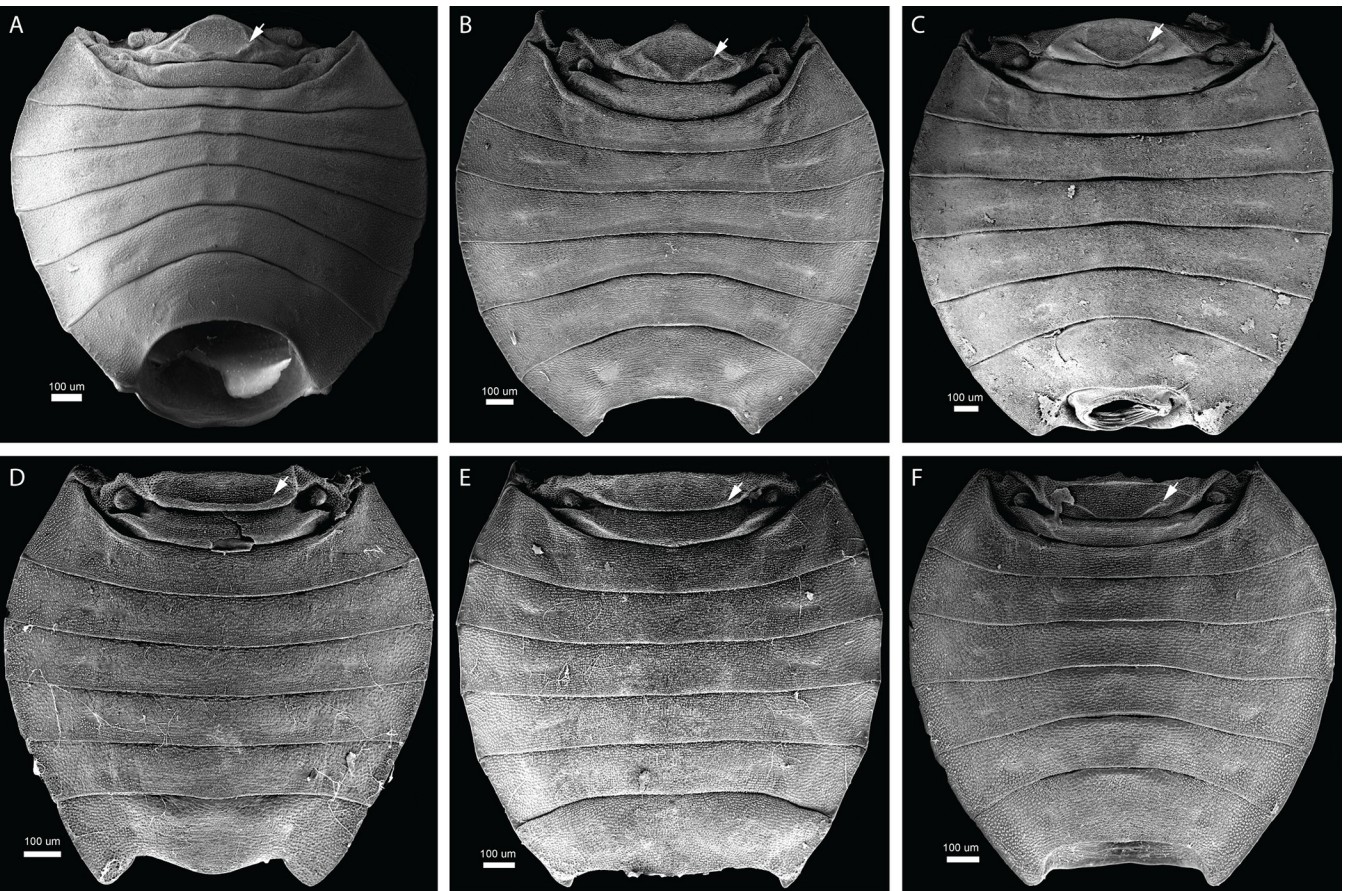

**Fig 4. Scanning electron microscopy images of the dorsal abdomen of a range of moss bugs, showing the morphological uniformity of the putative vibrational mechanism.** A) *Hackeriella brachycephala*; B) *Hemiodoecus leai*; C) *Peloridium hammoniorum*; D) *Oiophysa cumberi*; E) *Xenophyes kinlochensis*; F) *Xenophysella stewartensis*. The white arrow indicates the position of the posterolateral ridge.

muscles are the sheetlike VLMs (Figs 6D and 7A) and DLMs (Figs 6B and 7B). None of the muscles attach to the PLR (Figs 6B and 7B). The arrangement and proportion of the muscles operating the moss bug mechanism are more similar to the pregenital abdomen of many Heteroptera [55,57], which produce vibrational signals by means of simple abdominal tremulation [16,58].

The peloridiid pregenital abdomen does not display any noticeable sexual dimorphism (Fig 5), with only tergite I being slightly broader and with a thicker posterolateral ridge in females (Fig 5B). Acoustic signals have so far only been recorded from males [45,59]. The moss bug vibrational organ is probably functional only in the adult stage, as in nymphs, the first two abdominal pregenital segments are undeveloped, largely fused and probably immobile, both dorsally and ventrally (Fig 5C and 5D). Also, nymphs of several species were investigated via laser vibrometry, yet they did not produce any signals.

## Bioacoustics and biomechanics of the moss bug vibrational mechanism

Records of *Peloridium hammoniorum* Breddin, 1897, *Peloridium pomponorum* Shcherbakov, 2014 and *Xenophyes cascus* Bergroth, 1924 were produced with laser vibrometry. For the two *Peloridium*, video recordings of singing males could also be made (S1 Video). The features of the songs and generalized information on Peloridiidae are to be summarized elsewhere

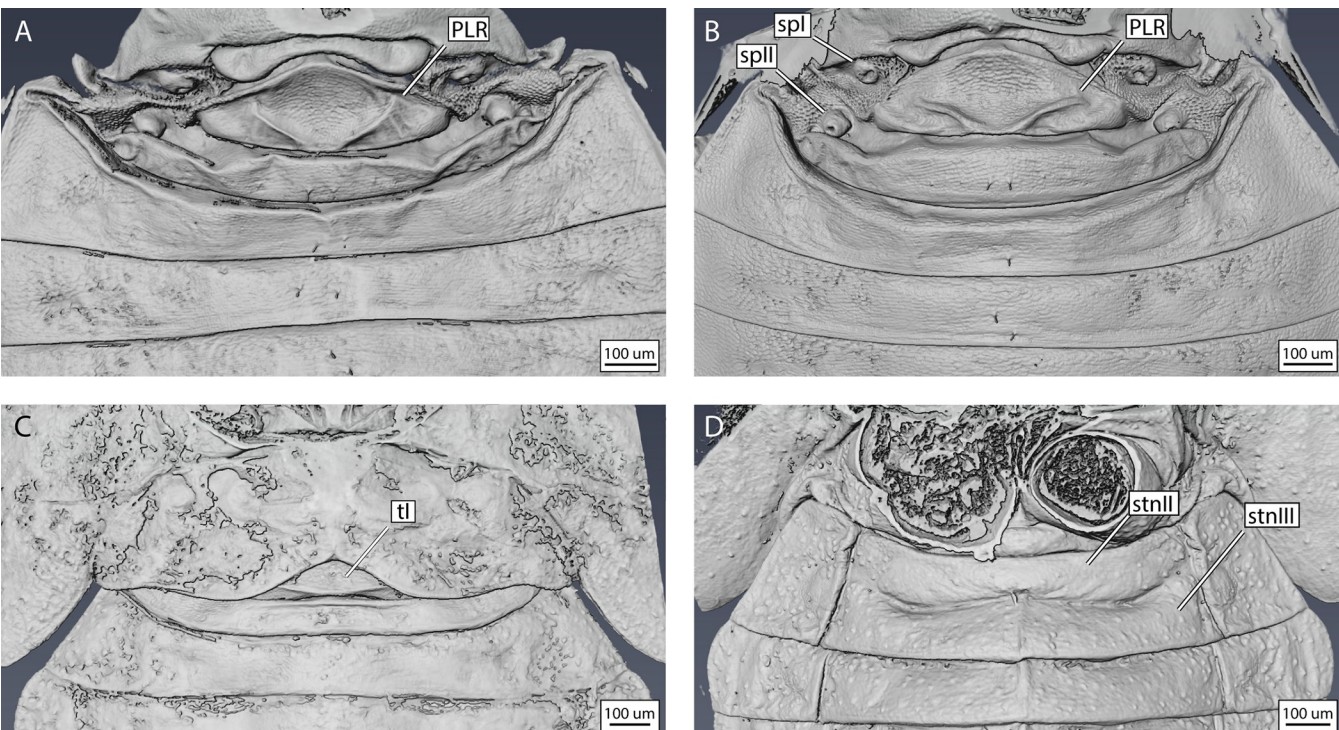

**Fig 5. The moss bug vibrational organ in different sexes and life stages of the species *Hemiodoecus leai*, as visualised by a three dimensional (3D) volumetric reconstruction of synchrotron X-ray microtomography images.** A) Male, dorsal view; B) female, dorsal view; C) last instar nymph, dorsal view; D) same, ventral view. Abbreviations: PLR = posterolateral ridge; sp = spiracle;st = sternite; t = tergite.

(Hartung, in preparation); here we use the signal structure of *P. hammoniorum* (whose movements while singing are identical to *P. pomponorum*), our best studied species, to extract information on the morphology of the vibroacoustic apparatus.

The call of *Peloridium hammoniorum* is a simple monotonous sequence of pulses with a fundamental frequency of approximately 500–600 Hz, and a dominant frequency extending to 4–5 kHz (Fig 8A). Such frequencies are within the range of what can be achieved by simple abdominal tremulation [3,16–19,58], in the absence of complex elastic recoil devices. The call of *P. hammoniorum* can be subdivided into two distinct stages–a regular song and a climax song (Fig 8B). The main component of the regular song is a pulse of high fundamental frequency (500 Hz-4 kHz) that sometimes is preceded by a sequence of several pulses of low fundamental frequency (500–1000 Hz; Fig 8C and 8D). The pulses with the high frequency follow each other ca. every second. The climax song, which typically lasts only for about 10 seconds (Fig 8B), is essentially a sped-up version of the regular song (pulse frequency of 3–4 Hz), that lacks the intermediate low frequency pulses of the latter and consists only of high frequency pulses (Fig 8E). Although the dominant frequency of the high frequency pulses of the climax is the same with the those of the regular song, the fundamental frequency of the former is slightly lower (3 kHz; Fig 8F) than the latter. After the climax song, the animal either resumes the regular song, or stops singing altogether.

Analysis of video recordings of singing animals show that each vibrational cycle of the regular song begins with the abdomen in its relaxed position (Fig 9A; S1 Video). The abdomen is then raised very slightly (Fig 9B; S1 Video) and returns to its relaxed position again (Fig 9C; S1 Video). We suggest that these low amplitude motions are responsible for the low frequency pulses of the regular song (Fig 8B and 8C). At some point the abdomen is lifted considerably

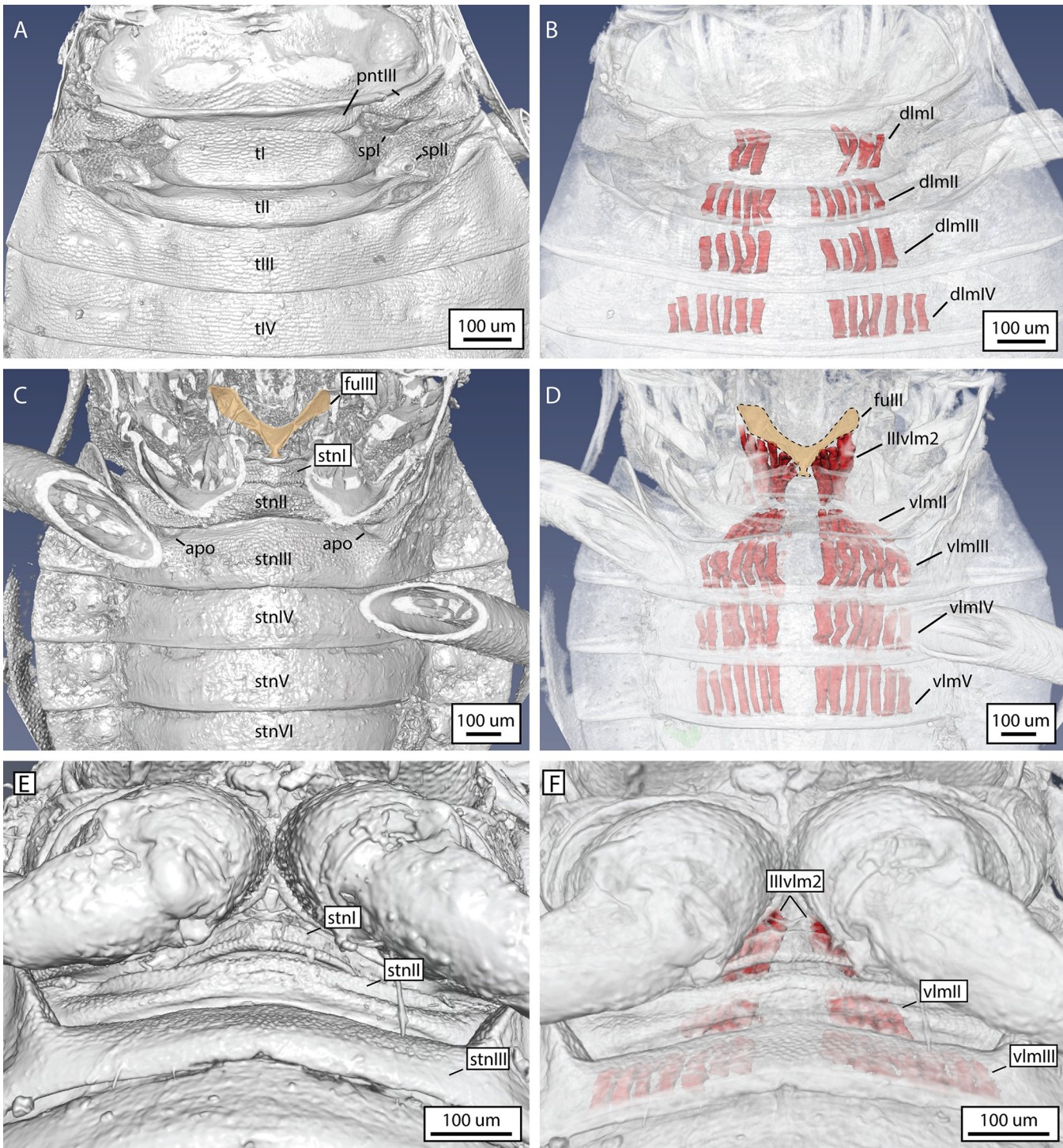

**Fig 6. Three dimensional (3D) volumetric reconstruction of exoskeleton and part of the musculature of the vibrational organ of the moss bug** *Xenophyes cascus*, **based on images generated by synchrotron X-ray microtomography.** A) Dorsal view; B) same, showing the dorsal longitudinal muscles (dlms); C) ventral view of pregenital abdomen, with an emphasis on the robust metafurca (fu3, in yellow); D) same, showing the ventral longitudinal muscles (vlms), focussing on the enlarged IIIvlm2 muscle that attaches on the metafurca; E) ventral view of proximal pregenital abdomen, showing the presence of a ring-like sternite one (stn1); F) same, showing the ventral longitudinal musculature of the region. Abbreviations: dlm = dorsal longitudinal muscle; pnt = postnotum; sp = spiracle; t = tergite; stn = sternite; vlm = ventral longitudinal muscle.

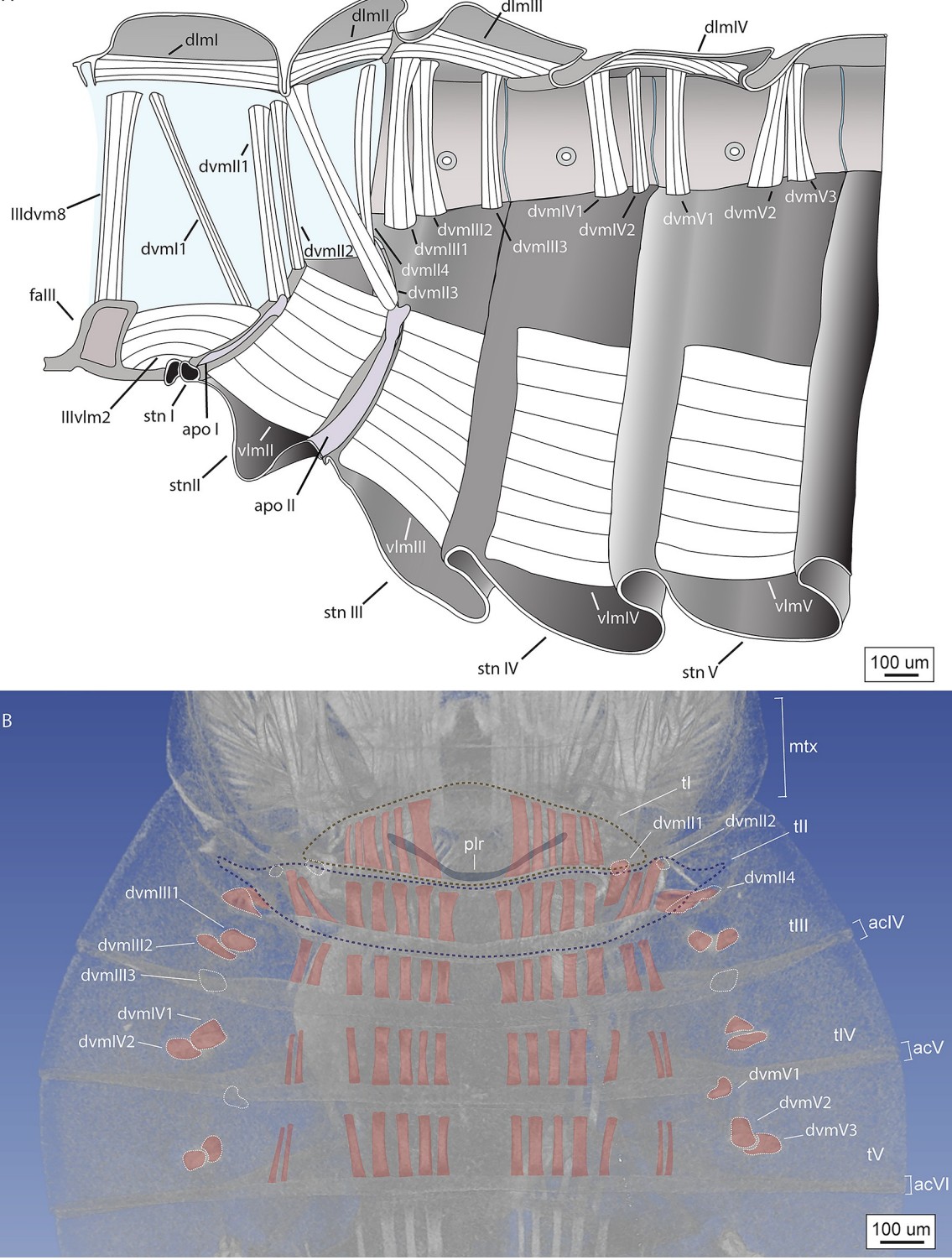

**Fig 7. Reconstruction of skeletonmuscular system of the moss bug vibrational organ.** A) Schematic illustration of the skeletonmusculature of a generalized moss bug (based on sections and X-ray microtomography of several species); B) dorsal musculature of *Peloridium hammoniorum*, 3D volumetric reconstruction of benchtop X-ray microtomography images. White dashed lines indicate the outline of muscles that are not preserved in this particular specimen, but were observed in sections of this species, and in X-ray images of related species. The outline of the posterolateral ridge was traced from the external surface of the 3D volumetric reconstruction. Abbreviations: apo = apodeme; dlm = dorsal longitudinal muscle; dvm = dorsoventral muscle; plr = posterolateral ridge; stn = sternite; t = tergite; vlm = ventral longitudinal muscle.

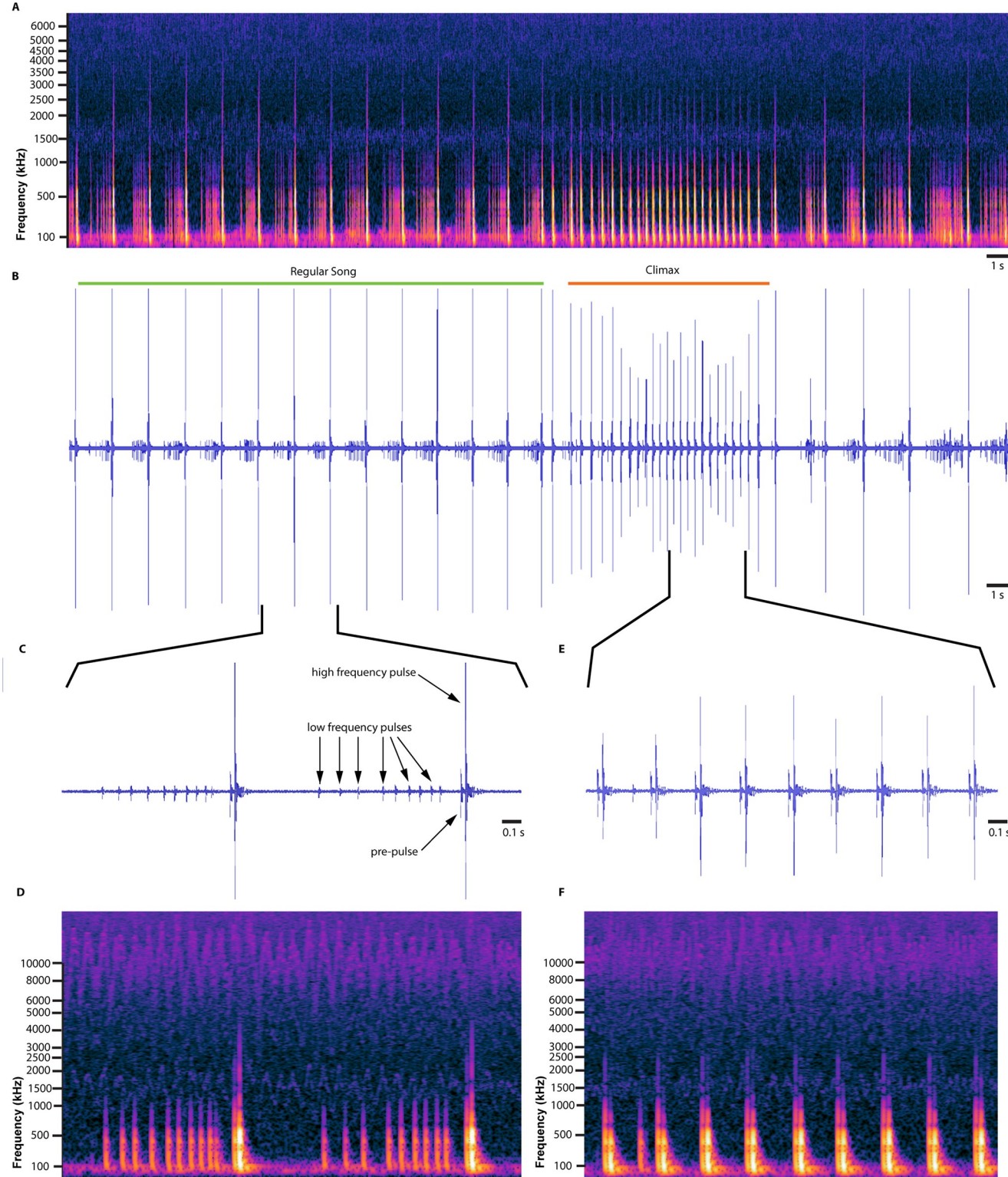

**Fig 8. The call of the South American moss bug *Peloridium hammoniorum*, recorded using laser Doppler vibrometry.** A) Spectrogram; B) wave form of courtship call, its two distinct phases–the regular song, and the climax song; C) snippet of two pulses of the regular song, with arrows emphasizing the series of low frequency pulses that precede the high frequency pulse; D) spectrogram of the same snippet; E) snippet of nine pulses of the climax song, showing that it comprises entirely of pre-pulses and high frequency pulses; F) spectrogram of the same snippet.

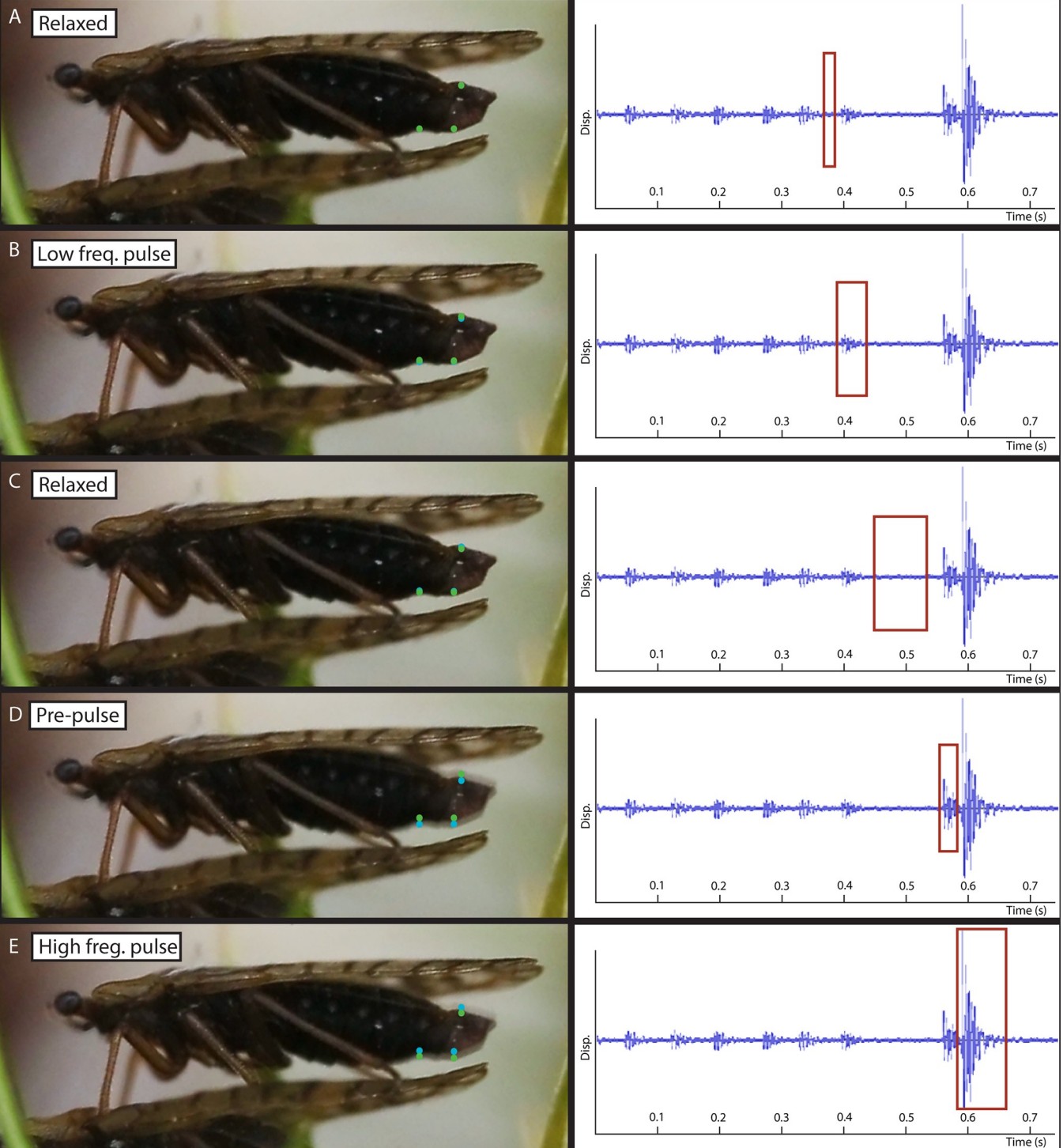

**Fig 9.** Stages of abdominal tremulation of a male *Peloridium hammoniorum* (left) and their corresponding stages in the waveform of the courtship song (right): (A) relaxed, (B) low frequency pulse, (C) relaxed, (D) pre-pulse; (E) high frequency pulse (end of cycle). Green and blue circles indicate position of other selected areas of the snapping organ in the current and previous panel, respectively. The red box on the waveform panel indicates vibrational activity associated with the stage of motion represented in that panel. The displacement axis is the same in all panels.

higher than before (Fig 9D), whose movement corresponds to the distinct pre-pulse prior to each high frequency pulse (Fig 9D). The high frequency pulse is generated once the abdomen rapidly returns to its original relaxed position (Fig 9E), and the song cycle is completed.

Although we do not have direct recordings of muscle activity, the only muscles that are anatomically capable of lifting the abdomen upwards are certain DVMs (IIIdvm8) and DLMs (dlmI), while the downwards motion is caused by DVM relaxation and contraction of VLMs, (IIIvlm2, vlmII), which are the primary retractors of the abdomen [60–62]. This function is further supported by the presence of apodemes I-II on the abdominal sterna, which serve as attachment sites for the VLMs (IIIvlm2, vlmII) (Fig 7A), suggesting that these regions undergo considerable mechanical stress. The differences in amplitude between the low frequency and high frequency pulses may be accomplished by different degrees of contraction of the three muscles involved, or differential contraction in the levators (only IIIdvm8 or dlmI contracts in low frequency pulse, while both contract concurrently during the high frequency pulse). If our interpretation of the mechanism is accurate, then the climax song is achieved solely by the high amplitude abdominal motions that we propose are responsible for the high frequency peaks of the regular song (Fig 9D). Directing the laser on the moss bugs' body in an attempt to measure possible contractions of muscles or changing geometry of sclerites caused the animals to stop singing and triggered an escape response, which prevented us from obtaining data on the energetics of vibration production in these insects.

Overall, neither the internal and external morphology of moss bugs nor their vibrational song suggest the presence of an elastic recoil device, indicating that simple abdominal tremulation is the underlying mechanism. Hence, referring to the mechanism of Coleorrhyncha as a tymbal (a ribbed, buckling structure, primarily operated by DVMs) [23,45] is morphologically and functionally imprecise. The biomechanical role of the posterolateral ridge (PLR) of tergite 1 remains poorly understood. The function of a spring that brings the abdomen back into position is unlikely since the PLR is not sclerotised, nor do muscles attach directly onto it (Fig 7B), and it does not display blue fluorescence under CLSM (Fig 3B and 3D), as high-stress cuticle with resilin does [20–22]. We suggest that a more likely function of the PLR is that of a point of weakness that enables successful deformation of tergite one upon contraction of muscle dlmI.

## Deep time morphological conservatism of the moss bug vibrational mechanism

We next sought to examine the pregenital abdominal morphology in extinct relatives of moss bugs. We focused on two exceptionally preserved fossils, *Karabasia evansi* Popov & Scherbakov, 1991 (Karabasiidae) and *Hoploridium dollingi* Popov & Scherbakov, 1991 (Hoploridiidae), dated to the Late Jurassic (circa 160–145 mya) and the Early Cretaceous (145–100 mya), respectively. Recent phylogenies suggest that Hoploridiidae is the sister group to Peloridiidae, whereas Karabasiidae was recovered as sister to the Hoploridiidae-Peloridiidae clade [33], even though these proposed affinities are in need of further study [34]. Our examination revealed remarkable morphological conservatism in the abdomen of Coleorrhyncha.

In *K. evansi* the dorsal surface of the abdomen (Fig 10A and 10B) is nearly identical to that of *Peloridium* (Fig 10C), including the long but rounded tergite one, a key element of the vibrational mechanism (as it contains the DLMs, the principal levators of the abdomen), and the strip-like tergite two. As for *H. dollingi*, its dorsal abdominal morphology is not preserved, but ventral structures (Fig 10D and 10E) do not differ from that of extant Coleorrhyncha (here exemplified by *Hemiodoecus leai*, Fig 10F), particularly in the morphology of the laterotergites and the position of the spiracles. We suggest that *H. dollingi*'s dorsal structure of abdomen is unlikely to differ from that of extant Coleorrhyncha. Details of the internal musculature are

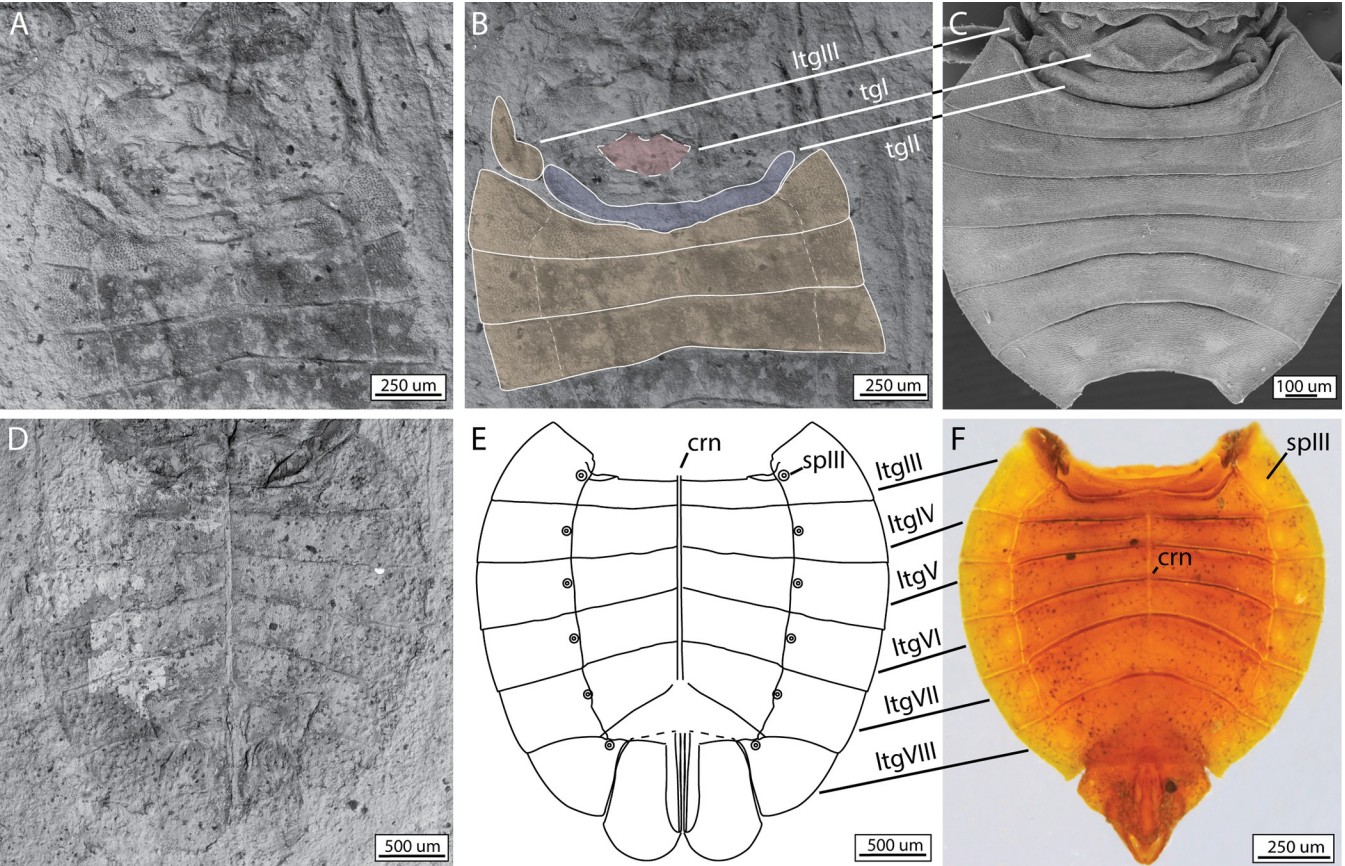

**Fig 10. Comparison of the pregenital abdominal morphology of extinct moss bug relatives to that of extant species.** A) Ventral surface of *Karabasia evansi* (holotype male, 3015/380), with dorsal structures (tergites I-II) visible, imaged with scanning electron miscroscopy; B) same, with an illustrated interpretation of its morphology; C) scanning electron microscopy image of the dorsal pregenital abdomen of the extant species *Hemiodoecus leai*; D) ventral surface of *Hoploridium dollingi* (holotype female 1989/3555), imaged with scanning electron miscroscopy; E) same, with a schematic reconstruction of its morphology and how it relates to; F) the ventral abdominal morphology of the extant species *H. leai*. Note that the holotype of *H. dollingi* is a female, whereas the extant specimen of *H. leai* is male. The pregenital abdominal morphology is largely identical in both sexes. Abbreviations: crn = median carina; ltg = laterotergite; sp = spiracle; t = tergite. (Photographs of the fossils courtesy Roman Rakitov and Dmitry Shcherbakov).

also not preserved in these fossils, nor is it possible to reconstruct the biomechanics of vibration production. However, based on the data presented here, it is evident that the gross abdominal structure of Coleorrhyncha, including those parts involved in vibrational signal production (tergites I-II) in modern forms, has remained largely unchanged for more than 145 million years. It remains to be found whether presumed stem-Coleorrhyncha, such as the family Progonocimicidae, possessed the same vibrational mechanism. We were unable to find fossil Progonocimicidae where the basal portion of the pregenital abdomen is visible, while their affinities to peloridiids are considered doubtful by some authors [34].

## Phylogenetic analysis

We next sought to incorporate the newly described morphological characters of the peloridiid vibrational mechanism into a previously used morphological matrix that comprised 93 characters observed with Scanning Electron Microscopy (SEM) [59] (S1 File). Two phylogenetic hypotheses were produced–one with 18 characters of the pregenital abdomen (Fig 11A; S2 File), and one with the latter integrated with the morphological matrix used in [59] (Fig 11B; S3 File). All morphological characters used are summarised in S4 File.

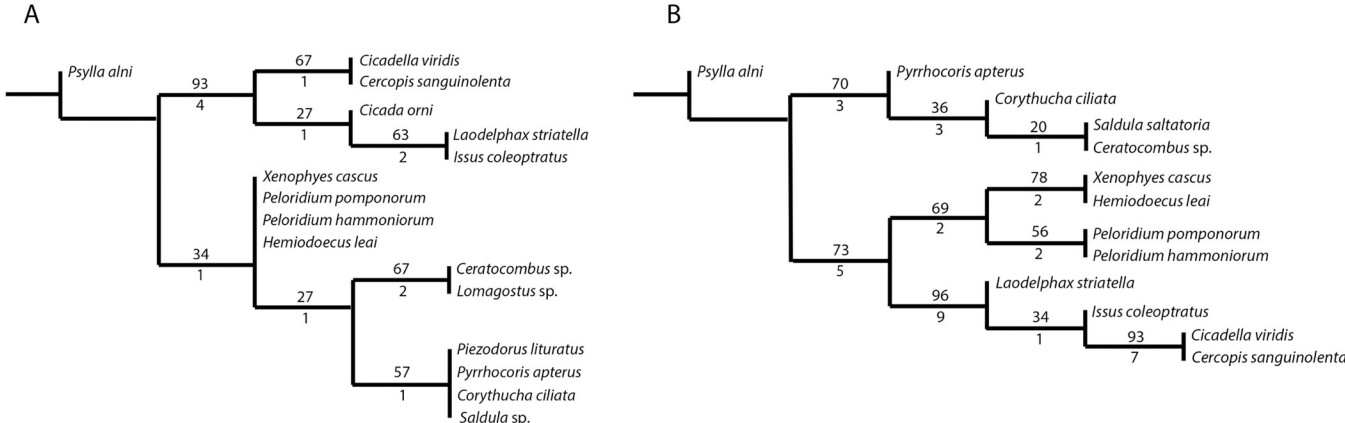

**Fig 11. Results of maximum parsimony phylogenetic analyses performed in TNT on different sets of characters.** Numbers above the nodes represent bootstrap values, below–the Bremer support values. A: 18 characters of the pregenital abdomen pertaining to the putative vibroacoustic apparatus; B: characters from matrix A integrated into the matrix of 93 morphological characters used in [59], involving only the taxa used in both [59] and this study.

The traditional search of the most parsimonious trees on a matrix including only the characters of the pregenital abdomen (S2 File) resulted in a single most parsimonious tree (9.605 rearrangements, best score 24, best score hit 10 times out of 10). In the resulting tree, Auchenorrnyncha and Heteroptera were retained as monophyletic, although Heteroptera with only 27 in bootstrap and 1 in Bremer support value (Fig 11A). The four peloridiid species remained in the polytomy, but formed a monophyletic group together with Heteroptera–although here as well, with only 34 bootstrap and 1 Bremer support value (Fig 11A).

The integrated set (Fig 11B) with the characters of the pregenital abdomen and the previously used SEM characters (S3 File), when analysed in traditional search with the same parameters, resulted also in a single most parsimonious tree (7.623 rearrangements, best score 167, hit 10 times out of 10) (Fig 11B). In this tree, Heteroptera, Auchenorrhyncha and Peloridiidae are all recovered as monophyletic (Fig 11B), with Peloridiidae having the least support (69 bootstrap, 2 Bremer support). As in the analysis by [59] (S1 File), performed exclusively with SEM characters, Peloridiidae was recovered as the sister group of Auchenorrhyncha, with a bootstrap value of 75 and Bremer support of 5 (Fig 11B).

## Discussion

In the present work, we attempted to elucidate the evolution of complex vibroacoustic mechanisms in Hemiptera. To this end, we revealed the previously unknown functional morphology of the evolutionarily important and biomechanically unstudied moss bugs, and we studied the bearing of this new morphological information for their phylogenetic relationships to groups of other Hemiptera. We find that the inclusion of vibrational organs in phylogenetic matrices may be a useful approach for testing existing hypotheses on hemipteran systematics. Some new phylogenomic studies indicated that Peloridiidae are the sister group to Auchenorrhyncha [52,54]. The characters of the pregenital abdomen and putative vibrational organ, when considered alone, support a closer relatedness to the true bugs (Fig 11A), in accordance with the more traditional phylogeny of Hemiptera [51], although the statistical support was quite poor in this case (Fig 11A). However, when the morphological matrix comprising abdominal characters is merged with a previously published matrix [59], Peloridiidae are recovered as forming a monophyletic group with Auchenorrhyncha (Fig 11B), in agreement with the latest phylogenomic studies [52,54].

Regardless of topology, the most parsimonious hypothesis is that the moss bugs ancestrally lack elastic recoil devices, the latter evolving only in Auchenorrhyncha (whether the snapping organ of planthoppers and the tymbals of cicadas represent homologous or independently evolved mechanisms remains to be proven [23]). Although we cannot exclude that Peloridiidae ancestrally possessed elastic recoil devices but lost them secondarily, this is highly unlikely. The abdomen of putative extinct relatives of peloridiids from the Late Jurassic-Early Cretaceous displays remarkable morphological similarity to that of modern moss bugs, suggesting that abdominal tremulation is the ancestral vibrational mechanism for the principal groups of Coleorrhyncha, as is the case in Heteroptera [55,63]. Similarly, Furthermore, loss of elastic recoil devices in Hemiptera is remarkably rare, represented by only five instances of tymbal-less species of cicadas [64–66]. Even in this case, however, at least two of the tymbal-less cicadas compensate for this loss by producing acoustic sound through percussion [66], thereby showing the behavioural significance of vibroacoustic organs and elastic recoil devices in the Auchenorrhyncha.

Based on the above, we suggest that simple abdominal tremulation likely first appeared in the common ancestor of Euhemiptera, which is dated to the Carboniferous (350 mya [52]), in accordance with previous studies [6,16,55,63,67]. Tremulation may therefore represent one of the oldest and most widely distributed modes of vibrational signalling in the animal kingdom.

If the above hypothesis is correct, the vexing question is to determine the selective pressures that led to the evolution of elastic recoil devices in Auchenorrhyncha. Elastic recoil mechanisms allow microscopic animals to overcome the limitations of their size and enable long-range transmission of sensorially efficient vibrational signals [68,69]. However, generating vibroacoustic signals with elastic recoil devices may be energetically costly [70], while the sheer anatomical complexity of tymbals and snapping organs [21,22] may require a series of morphological transformations that are developmentally challenging to achieve. Although hemipteran abdominal tremulation may represent a biomechanically efficient means of generating vibrational signals for more than 350 million years, the emitted calls are rather uniform (Fig 8; [4,6]). Insects singing in sexual contexts are under strong selective pressure to enhance their vibroacoustic repertoire with more complex songs [71], yet abdominal tremulation might be limited in the number of distinct signals that it can produce [16]. To overcome this constraint, Hemiptera supplement their tremulatory courtship songs with percussion [17,18], wing buzzing [19], and an extremely diverse set of stridulatory mechanisms [16]. Likewise, we propose that elastic recoil devices such as the snapping organ and the tymbal may have arisen in Auchenorrhyncha due to intense sexual selection for more complex courtship signals, which would have been impossible to achieve by simple abdominal tremulation alone.

In conclusion, our study addresses one of the last outstanding knowledge gaps in hemipteran vibroacoustic signalling evolution. At the same time, we provide novel morphological and generalized data that offer themselves for further hypothesis testing on the behavioural, physical, and developmental drivers that led to the diversity of hemipteran vibroacoustic mechanisms that we observe today. Understanding the developmental pathways that lead to the development of hemipteran elastic recoil devices will be essential towards resolving their homologies.

## Material and methods

### Field work

Peloridiidae specimens were collected in Australia in 2009–10, in New Zealand 2010 and in Chile 2014 –bryophyte samples were first sifted and then the sifted material was analyzed for up to 24 hours (depending on the moisture) in Berlese funnels. Collections in Australia were

done under permits WITK06355209 (Queensland), S13005 (New South Wales), 10005138 (Victoria) and FA10018 (Tasmania); in New Zealand under WE-26346-RES; in Chile under a permit issued by CONAF.

## Bioacoustic recordings

Acoustic signals of Peloridiidae were recorded with the Polytec PDV100 vibrometer and a Roland digital recorder (sampling rate: 44100 Hz, bit depth: 16 bit). Recordings were made in closed plastic vials with wet bryophytes with the peloridiid specimens sitting on them. The laser beam of the vibrometer was directed at a small piece of reflector foil glued to the stem of the bryophyte where the specimens were sitting. Oscillograms were analyzed with Audacity 2.1.3 and spectrograms were constructed using Raven Lite 2.0 (Cornell Lab of Ornithology).

## Confocal Laser Scanning Microscopy (CLSM)

Specimens of *Hemiodoecus leai* and *Xenophyes cascus* were placed between two cover slips in 70% ethanol. Images were taken with an Olympus FV1000, at a laser wavelength of 488 nm.

## Scanning electron microscopy (SEM)

All specimens of Peloridiidae were first manually cleaned using a paintbrush and fine needle after overnight incubation in ethyl acetate. The specimens were then critically-point dried on a BAL-TEC CPD 030, sputtercoated on Quorum SC7640 and Quorum Q150RS. Scanning electron microscopy was performed on a Zeiss EVO LS10. Fossil Coleorrhyncha were imaged by Roman Rakitov with a Tescan Vega3 scanning electron microscope at the Paleontological Institute, Russian Academy of Sciences.

## Microcomputed tomography (micro-CT)

Three *H. leai* and one *X. cascus* ethanol-preserved specimens were scanned at the TOMCAT beamline, Swiss Light Source (SLS), Paul Scherrer Institut, Switzerland, at a beam energy of 16 keV with final pixel size of 0.65 um. Additionally, two ethanol-preserved specimens of *P. hammoniorum* and *P. pomponorum* were subjected to micro-tomographic analysis at the Museum für Naturkunde Berlin (SCR_022585) using a Phoenix nanotom X-ray tube (Waygate Technologies, Baker Hughes, Wunstorf, Germany; SCR_022582) at 70kV and 150 or 200μA, generating 1000 projections with 750ms per scan. The effective voxel size was 4,2 um. The cone beam reconstruction was performed using the datos|x 2 reconstruction software (Waygate Technologies, Baker Hughes, Wunstorf, Germany; datos|x 2.2) and the data were visualized in VG Studio Max 3.5 (Volume Graphics GmbH, Heidelberg Germany).

Three-dimensional reconstruction of both the synchrotron and benchtop micro-CT scans was undertaken using Amira 6.1 software (Mercury Systems).

## Photomicrography

A male ethanol-preserved specimen of *H. leai* was imaged with a Leica M165c microscope equipped with a Leica DFC490 camera. The resulting stacked images were combined using Helicon Focus (Helicon Soft, Kharkiv, Ukraine) or VHX-5000 system software.

## Videography

Records of two Chilean specimens of *P. hammoniorum* were performed on native *Polytrichadelphus* moss at room temperature, using a Canon 5D Mark II camera (30 frames per second), MP-E 65mm lens and Macro Ring Lite MR-14EX cold light source.

## Phylogenetic analysis

Character matrices for phylogenetic analysis were produced with WinClada 1.00.08 [72]. The 18 abdominal characters were analysed either on their own (S2 File), or together with the matrix of 93 morphological characters established with SEM [59] (S3 File).

Phylogenetic analysis was performed with TNT version 1.1 (sponsored by the Willi Hennig Society, [73]), one of the most efficient packages utilizing maximum parsimony methods. *Psylla alni* was set as the root; characters were unweighted and non-additive. Traditional search was performed with default settings (Wagner trees, 1 random tree, 10 replications, TBR (tree bisection reconnection) swapping algorithm, 10 trees to save per replication). Every time a new traditional search was performed, the program was started anew to avoid influence of trees that could be stored in the buffer.

Bremer support values were counted by TBR from existing trees, retaining trees suboptimal by 20 steps. Images provided by TNT were enhanced using Adobe® Illustrator® 2021 and Adobe® Photoshop® 2021.

## Supporting information

**S1 Table. A list of all the moss bug species examined and their label information.**
(DOCX)

**S2 Table. List of the muscles associated with the moss bug vibrational organ and their attachments.**
(XLSX)

**S1 Video. Videography of a male *Peloridium hammoniorum* generating its call.**
(MP4)

**S1 File. Data matrix with 93 characters of microscopic morphologies obtained with SEM [59].**
(SS)

**S2 File. Data matrix with 18 characters of the pregenital abdomen, obtained in this study, used to produce the phylogenetic hypothesis in Fig 11A.**
(SS)

**S3 File. An integrated dataset combining the matrices in S1 and S2 Files, with only those taxa retained whose character states were known for both matrices, used to produce the phylogenetic hypothesis in Fig 11B.**
(SS)

**S4 File. List of all morphological characters used in phylogenetic analysis.**
(DOCX)

## Acknowledgments

The authors thank Kristin Mahlow (Berlin Museum of Natural History) for obtaining micro-CT scans of *P. hammoniorum* and *P. pomponorum*. The authors acknowledge the Paul Scherrer Institut, Villigen, Switzerland for provision of synchrotron radiation beamtime at beamline TOMCAT X02DA of the SLS. Beamtime was granted at PSI under a project co-proposed by Davranoglou L.-R., Beth Mortimer, and Graham K. Taylor (University of Oxford). The authors are grateful to Dávid Rédei for his helpful comments on peloridiid morphology, and to Dimitri Scherbakov and Roman Rakitov for providing the scanning electron images of the

fossil specimens and informative discussions of the fossils' morphology. Jürgen Deckert kindly provided video and photo materials of Peloridiidae. André Nel provided rare papers on fossil Coleorrhyncha. Nature protection officials in Australia, Chile and New Zealand provided collection and export permits (Material and Methods).

## Author Contributions

**Conceptualization:** Leonidas-Romanos Davranoglou, Viktor Hartung.

**Data curation:** Leonidas-Romanos Davranoglou, Viktor Hartung.

**Formal analysis:** Leonidas-Romanos Davranoglou, Viktor Hartung.

**Funding acquisition:** Leonidas-Romanos Davranoglou, Viktor Hartung.

**Investigation:** Leonidas-Romanos Davranoglou, Viktor Hartung.

**Methodology:** Leonidas-Romanos Davranoglou, Viktor Hartung.

**Project administration:** Leonidas-Romanos Davranoglou, Viktor Hartung.

**Resources:** Leonidas-Romanos Davranoglou.

**Software:** Leonidas-Romanos Davranoglou, Viktor Hartung.

**Validation:** Leonidas-Romanos Davranoglou, Viktor Hartung.

**Visualization:** Leonidas-Romanos Davranoglou, Viktor Hartung.

**Writing – original draft:** Leonidas-Romanos Davranoglou.

**Writing – review & editing:** Leonidas-Romanos Davranoglou, Viktor Hartung.

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
