## [Decision Letter · Decision Letter 0]

2 Jan 2024

PONE-D-23-39772Moss bugs shed light on the evolution of complex bioacoustic systemsPLOS ONE

Dear Dr. Davranoglou,

Thank you for submitting your manuscript to PLOS ONE. After careful consideration, we feel that it has merit but does not fully meet PLOS ONE’s publication criteria as it currently stands. Therefore, we invite you to submit a revised version of the manuscript that addresses the points raised during the review process.

We look forward to receiving your revised manuscript.

Kind regards,

Feng ZHANG, Ph.D.

Academic Editor

PLOS ONE

Journal Requirements:

"Leverhulme Trust Early Career Fellowship grant (ECF-2021-199); Elsa Neuman doctorate grant (application number H49023); German Academic Exchange travel grant (processing number D/09/04219)"

3. Thank you for stating the following in the Acknowledgments Section of your manuscript:"This publication arises from research funded by the Leverhulme Trust Early Career Fellowship grant (ECF-2021-199) and the John Fell Oxford University Press Research Fund to L.-R.Davranoglou. V. Hartung was funded by the Elsa Neuman doctorate grant (application numbe H49023) and the German Academic Exchange travel grant (processing number D/09/04219)."

Please remove any funding-related text from the manuscript and let us know how you would like to update your Funding Statement. Currently, your Funding Statement reads as follows: "Leverhulme Trust Early Career Fellowship grant (ECF-2021-199); Elsa Neuman doctorate grant (application number H49023); German Academic Exchange travel grant (processing number D/09/04219)"

4. We notice that your supplementary tables are included in the manuscript file. Please remove them and upload them with the file type 'Supporting Information'. Please ensure that each Supporting Information file has a legend listed in the manuscript after the references list.

Reviewers' comments:

Reviewer's Responses to Questions

**Comments to the Author**

1. Is the manuscript technically sound, and do the data support the conclusions?

Reviewer #1: Yes

Reviewer #2: Yes

2. Has the statistical analysis been performed appropriately and rigorously? 

Reviewer #1: N/A

Reviewer #2: N/A

3. Have the authors made all data underlying the findings in their manuscript fully available?

Reviewer #1: Yes

Reviewer #2: No

4. Is the manuscript presented in an intelligible fashion and written in standard English?

Reviewer #1: Yes

Reviewer #2: Yes

5. Review Comments to the Author

Reviewer #1: The author is obviously a highly skilled morphologist and a very competent expert in the field of insect bioacoustics. The study addresses vibrational organs of a phylogenetically crucial taxon in the megadiverse Hemiptera.

l. 23. I would avoid “derived Hemiptera”, strictly speaking there are only derived characters/character states. I assume you mean Euhemiptera?

l. 138: I recommend using Roman numerals for abdominal segments, tergites and sternites (like e.g. in Beutel et al. 2014) (e.g. tergite I).

l. 149. I do not really like “in most Heteroptera” (or “most Auchenorrhyncha”) but would rather use “most species/groups of….” Or “most heteropterans”, but this is disputable.

l. 180. Studies in prep. cannot be really cited, maybe use pers. comm. instead?

There are some comparative statements in the morphological description (referring to Auchenorrhyncha) but in this specific case I have no objections.

Phylogenetic analysis

I mostly worked with maximum parsimony myself, but apparently Bayesian inference can produce better results, also concerning ancestral state reconstruction. But the use of MI is optional in this case.

Literature: relevant studies are fully covered as far as I can see.

Language: very good

Illustrations: excellent

To summarize, this is an outstanding study, in terms of morphological documentation, vibration recording, phylogenetic evaluation, and evolutionary interpretation.

I recommend publication after (very) minor revision.

It was a pleasure to read this.

Rolf G. Beutel

Reviewer #2: The paper addresses the bioacoustic system in Hemiptera by analysing and comparing functional morphology, video films, and laser vibrometry from extant species of Peloridiidae (and ther fossil relatives) with other hemipteran bugs and thereby provide new insight in the origin of this important feature.

The used techniques are relatively easy to follow even for a reader with no personal experience in these methods, and the morphological characters are scored and used in phylogenetic analyses suggesting new evidence for the relationship between peloridiids and other hemipteran bugs, even though the support for such relationships is still inconclusive.

The paper is very well written, and I only have a few minor details that I would like to see changed.

1. It wonder why the newly generated morphological characters used for the phylogenetic analyses are not available and open for inspection and discussion.

2. While the paper reviews studies of the relationship between Peloridiids and other hemipteran bugs, the relationships within the family are not addressed, even though they could be used to discuss the difference between species from New Zealand (Oiophysa, Xenophyes, Xenophysella) on one side and species from Australia (Hackeriella, Hemiodoecus) and South America (Peloridium) on the other as outlined on lin 114-118. According to the recent study by Ye et al. (2019), Peloridium is sister group to all other peloridiids, and the fauna from New Zealand (and New Caledonia) is sister group to a clade of other South American taxa and another clade consisting of the species from Australia and Lord Howe Is.

3. On Fig. 1, it is quite confusing that two males of Hackeriella weitchi are shown on top of one another. It would make more sense to show a single male or a male and a female.

4. I wonder if the references for inferring the relationships between Peloridiidae and other hemipterans for Fig. 2 on line 83-86 are correct.

Other formalia:

• Do check if the right type of brackets are used (see. e.g. ref. 63 on l. 303-304.

• Reference 64 is first mentioned in line 350, which is after succeeding references, e.g. ref. 67 in line 346.

6. PLOS authors have the option to publish the peer review history of their article (what does this mean?). If published, this will include your full peer review and any attached files.

Reviewer #1: **Yes: **Rolf G. Beutel

Reviewer #2: No

---

## [Author Response · Author response to Decision Letter 0]

15 Jan 2024

See response to reviewers files attached with this submission.

---

## [Editor Report · Decision Letter 1]

22 Jan 2024

Moss bugs shed light on the evolution of complex bioacoustic systems

PONE-D-23-39772R1

Dear Dr. Davranoglou,

We’re pleased to inform you that your manuscript has been judged scientifically suitable for publication and will be formally accepted for publication once it meets all outstanding technical requirements.

Kind regards,

Feng ZHANG, Ph.D.

Academic Editor

PLOS ONE
---

## [Editor Report · Acceptance letter]

16 Feb 2024

PONE-D-23-39772R1 

PLOS ONE

Dear Dr. Davranoglou, 

I'm pleased to inform you that your manuscript has been deemed suitable for publication in PLOS ONE. Congratulations! Your manuscript is now being handed over to our production team.

Kind regards, 

on behalf of

Dr. Feng ZHANG 

Academic Editor

PLOS ONE